

# Characterization of shifts of koala (*Phascolarctos cinereus*) intestinal microbial communities associated with antibiotic treatment

Katherine E. Dahlhausen[1], Ladan Doroud[2], Alana J. Firl[1], Adam Polkinghorne[3] and Jonathan A. Eisen[1]

[1] Genome Center, University of California, Davis, CA, United States of America
[2] Department of Computer Science, University of California, Davis, CA, United States of America
[3] Centre for Animal Health Innovation, University of the Sunshine Coast, Sippy Downs, QLD, Australia

Corresponding author
Katherine E. Dahlhausen,
katdah@ucdavis.edu

## ABSTRACT

Koalas (*Phascolarctos cinereus*) are arboreal marsupials native to Australia that eat a specialized diet of almost exclusively eucalyptus leaves. Microbes in koala intestines are known to break down otherwise toxic compounds, such as tannins, in eucalyptus leaves. Infections by *Chlamydia*, obligate intracellular bacterial pathogens, are highly prevalent in koala populations. If animals with *Chlamydia* infections are received by wildlife hospitals, a range of antibiotics can be used to treat them. However, previous studies suggested that koalas can suffer adverse side effects during antibiotic treatment. This study aimed to use 16S rRNA gene sequences derived from koala feces to characterize the intestinal microbiome of koalas throughout antibiotic treatment and identify specific taxa associated with koala health after treatment. Although differences in the alpha diversity were observed in the intestinal flora between treated and untreated koalas and between koalas treated with different antibiotics, these differences were not statistically significant. The alpha diversity of microbial communities from koalas that lived through antibiotic treatment versus those who did not was significantly greater, however. Beta diversity analysis largely confirmed the latter observation, revealing that the overall communities were different between koalas on antibiotics that died versus those that survived or never received antibiotics. Using both machine learning and OTU (operational taxonomic unit) co-occurrence network analyses, we found that OTUs that are very closely related to *Lonepinella koalarum*, a known tannin degrader found by culture-based methods to be present in koala intestines, was correlated with a koala's health status. This is the first study to characterize the time course of effects of antibiotics on koala intestinal microbiomes. Our results suggest it may be useful to pursue alternative treatments for *Chlamydia* infections without the use of antibiotics or the development of *Chlamydia*-specific antimicrobial compounds that do not broadly affect microbial communities.

## BACKGROUND AND SIGNIFICANCE

The koala, *Phascolarctos cinereus*, is an arboreal marsupial native to Australia with multiple unique aspects to its biology. Joeys (baby koalas) live in their mother's pouch, relying on milk for nutrition for the first two months of life prior to switching to the consumption of pap for up to another six months. Pap is fecal matter excreted by the mother, which is more concentrated in nutrients and microbes than normal feces (*Osawa et al., 1995*). Pap consumption is an essential physiological activity for joeys as they transition to the adult koala diet consisting almost exclusively of eucalyptus leaves (*Osawa et al., 1995*; *Cork, Hume & Dawson, 1983*).

Eucalyptus leaves have high levels of tannins, soluble phenolic compounds that form complexes with proteins and are resistant to degradation, rendering them toxic to many species that eat them (*Aguilar et al., 2007*). It is assumed that koalas rely on tannin-degrading bacteria that colonize the koala intestines once a joey begins consuming pap from its mother (*Osawa et al., 1995*; *Goel et al., 2005*). Culture-based methods have revealed that two known tannin degrading types of microorganisms (*Streptococcus* sp. and *Lonepinella koalarum*) are found in the gastrointestinal tract of koalas (*Goel, Puniya & Singh, 2007*). Tannin-degrading bacteria are common amongst all animals with a high tannin diet, including koalas, and are thought to allow these animals to survive off of tannin-rich diets (*Kohl, Stengel & Dearing, 2015*; *Gasse, 2014*).

One factor contributing to the dramatic decline in koala populations is infection by bacteria in the *Chlamydia* genus, rates of which are as high as 100% in some koala populations (*Kollipara et al., 2013*). *Chlamydia* are Gram-negative intracellular bacterial pathogens, infecting a diversity of eukaryotic hosts including mammals, birds, reptiles, fish, and amoeba (*Bachmann, Polkinghorne & Timms, 2014*). Two species within the genus *Chlamydia* are known to infect koalas, *Chlamydia pecorum* and *Chlamydia pneumoniae*, with *C. pecorum* being the more prevalent and pathogenic of the two in this host (*Polkinghorne, Hanger & Timms, 2013*).

In koalas, *Chlamydia* infect the ocular site, urinary tract, and/or reproductive tract in both chronic and acute states (*Polkinghorne, Hanger & Timms, 2013*). Wild koalas with symptoms of *Chlamydia* infections, such as urinary incontinence or conjunctivitis, are routinely brought to wildlife hospitals to be tested and treated (*Gonzalez-Astudillo et al., 2017*; *Griffith et al., 2013*). Transmission of *Chlamydia* between koalas can be sexually transmitted and also through exposure to joeys when eating pap from an infected mother (*Polkinghorne, Hanger & Timms, 2013*). The treatment of *Chlamydia* infections in koalas is controversial. Although different antibiotics are routinely administered to koalas in care, several studies suggest antibiotic treatment has detrimental effects on koalas such as a severe loss in body weight, severe dysbiosis, and even death (*Polkinghorne, Hanger & Timms, 2013*; *Osawa & Carrick, 1990*; *Brown, 1987*). Most notably, a few studies suggest that although effective, *Chlamydia* infection treatment with the antibiotics chloramphenicol and enrofloxacin has adverse and even fatal side effects for koalas (*Osawa & Carrick, 1990*; *Black, Higgins & Govendir, 2015*; *Black et al., 2014*; *Govendir et al., 2012*; *Lawrence et al., 2016*). A possibly related finding is that rats lacking tannin-degrading bacteria while

eating a tannin-enriched diet had similar symptoms (e.g., decreased food intake) to those reported for koalas on antibiotic treatment (*Kohl, Stengel & Dearing, 2015*). A study of antibiotic sensitivity of two *Lonepinella koalarum* strains, known tannin-degraders, indicated sensitivity to chloramphenicol, the antibiotic that all antibiotic-treated koalas in this study were administered (*Osawa & Stackebrandt, 2015*) .

Despite the putative importance of koalas' intestinal microbes to their biology, only two culture-independent studies on the koala intestinal microbial community (microbiome) have been published, neither in the context of the impacts of antibiotics (*Alfano et al., 2015*; *Barker et al., 2013*). In the current study, we hypothesized that adverse side effects of antibiotics administered to koalas with *Chlamydia* infections are related to disturbance in the microbial communities present in the koala gastrointestinal tract. To test this hypothesis, we characterized the microbiome of koalas that either were treated or not treated with antibiotics over time. We then analyzed these microbiomes to examine how diversity patterns, individual taxa, and potential tannin degraders varied with respect to koala survival and antibiotic treatment.

## MATERIALS AND METHODS

This study was conducted in close collaboration with the Australia Zoo Wildlife Hospital (Queensland, Australia) and the Port Macquarie Koala Hospital (New South Wales, Australia). Sample collection from the Australia Zoo Wildlife Hospital and Port Macquarie Koala Hospital qualified for exemption from approval by the University of the Sunshine Coast Ethics Committee. Furthermore, appropriate permissions for transportation and use of samples for educational purposes was confirmed for export from the Australian Government and import from the United States Fish and Wildlife Service.

### Sample collection

Samples were collected from koalas admitted to the Australia Zoo Wildlife Hospital in Beerwah, Queensland, Australia and the Port Macquarie Koala Hospital, Port Macquarie, New South Wales, Australia. Koalas were lodged in outdoor, single-occupancy enclosures. They were fed a variety of fresh eucalyptus leaves that were collected daily from local forests. Standard disease transmission prevention procedures were used at each hospital. Only individuals that were sampled for at least 21 days were included in this study because we thought that given koalas' slow metabolisms and digestive mobility, we would still be able to capture immediate changes to the intestinal microbiome in this three-week period. However, this resulted in a limited number of control koalas because those that were not treated with antibiotics were released or died sooner than our 21-day cut off.

Seven koalas from the Australia Zoo Wildlife Hospital were included, six of which received the antibiotic chloramphenicol for the treatment of *Chlamydia* and one that did not receive any antibiotic treatment. Treatment involved a single, daily treatment of 60 mg/kg for varying durations; mL of the dosage is provided in Table 1. Four koalas were sampled from the Port Macquarie Koala Hospital, three of which were administered a combination of chloramphenicol and enrofloxacin antibiotics for the treatment of *Chlamydia* and one that did not receive any antibiotic treatment. Treatment involved a

**Table 1 Information on study cohort of koalas, treatments, and samples.** Table lists antibiotic used, dosage, fate and number of samples. Antibiotic treatment regime (type of antibiotic, dosage, and time course) and euthanasia decisions were determined by the assigned veterinarian for each koala.

| Koala ID | Antibiotic used | Daily dose | Lived or died | No. samples collected |
|---|---|---|---|---|
| A | Chloramphenicol | 2.6 ml Chloramphenicol | Lived | 30 |
| B | Chloramphenicol | 2.0 ml Chloramphenicol | Died | 7 |
| D | Chloramphenicol | 2.5 ml Chloramphenicol | Lived | 17 |
| E | Chloramphenicol | 2.1 ml Chloramphenicol | Lived | 9 |
| F | Chloramphenicol | 2.3 ml Chloramphenicol | Died | 9 |
| G | Chloramphenicol | 1.9 ml Chloramphenicol | Lived | 13 |
| H | No Antibiotics | 0 | Lived | 13 |
| J | Chloramphenicol and Enrofloxacin | 1.4 ml Enrofloxacin, 2.4 ml Chloramphenicol | Died | 10 |
| K | No Antibiotics | 0 | Lived | 14 |
| P | Chloramphenicol and Enrofloxacin | 1.3 ml Enrofloxacin, 2.6 ml Chloramphenicol | Lived | 12 |
| R | Chloramphenicol and Enrofloxacin | 1.1 ml Enrofloxacin, 2.4 ml Chloramphenicol | Lived | 8 |

single, daily dose of 10 mg/kg of enrofloxacin for 2 weeks, followed by a single, daily dose of 60 mg/kg of chloramphenicol for varying durations (Table 1).

Two animals from Australia Zoo Wildlife Hospital (antibiotic-treated) and one Port Macquarie Koala Hospital animal (also antibiotic-treated) were euthanized during the course of this study. These animals exhibited too poor of a health status that they were determined to be unfit for survival by their respective veterinarians.

A total of 141 fecal samples were collected for analysis in this study from the eleven animals receiving care. These samples were composed of fresh fecal pellets (<15 min since excretion) collected every three days, beginning on the day each koala was admitted (i.e., immediately prior to first dosage of antibiotics) through to the day each koala was either released back to the wild or deceased. Fecal material was collected from the floor or branch with gloved hands and sealed in a small, sterile plastic bag. Samples were transported to a lab at the University of the Sunshine Coast on ice (<1 h) from the zoo and stored at −80 °C (Table 1).

Additionally, 17 built environment samples were collected from individuals' enclosures. Samples were collected by a 10-second dry swabbing of floors and branches with a gloved hand, and sealed in a small, sterile plastic bag. Leaf samples were collected as well by removing a full leaves from the branches provided for each enclosure and sealing them in a small plastic bag. Samples were transported to a lab at the University of the Sunshine Coast on ice (<1 hr) from the zoo, and stored at −80 °C. These samples are included in Fig. 1, but were otherwise not analyzed in this report.

## DNA extraction

Based on examination of previous work, we concluded that fecal pellets were the best possible proxy for the overall intestinal microbial community without the use of invasive

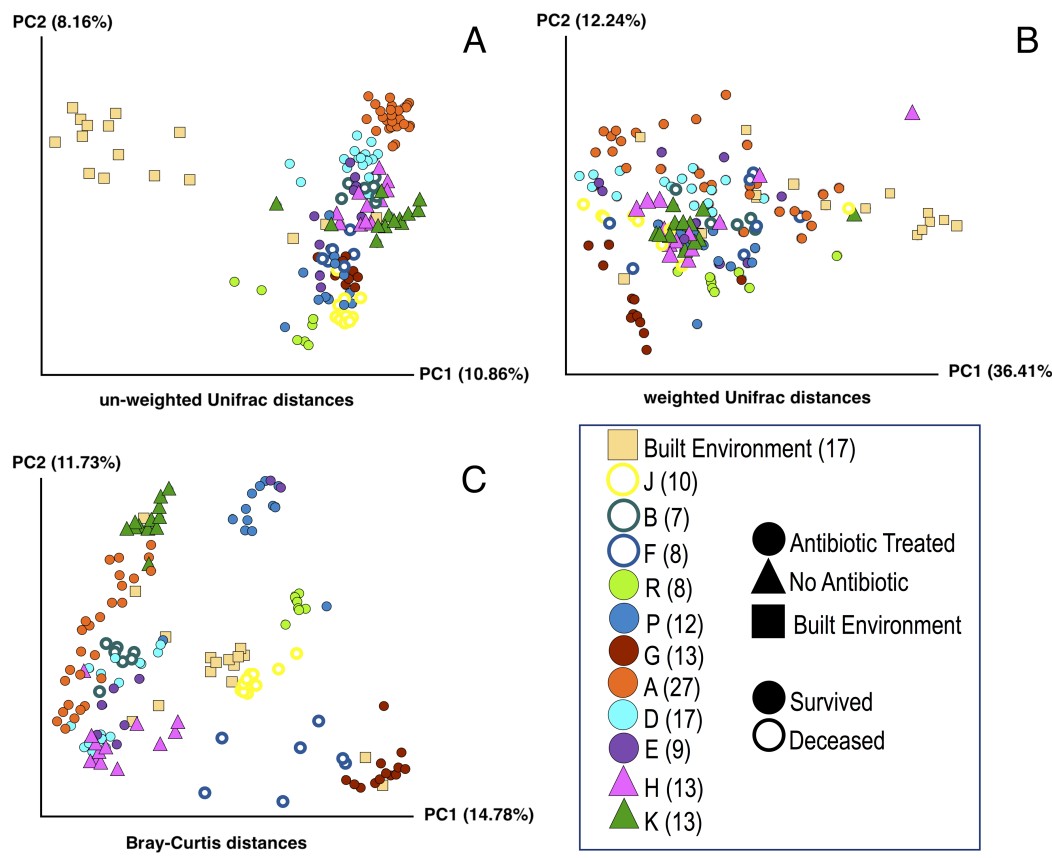

**Figure 1** **Principle Coordinate Analysis (PCoA) of microbial communities found in koala fecal samples based on analysis of rRNA gene sequences.** PCoA plots were generated using the QIIME (Quantitative Insights Into Microbial Ecology) version 1.9.1 workflow on sequencing reads following quality filtering and chimera removal. PCoA plots are shown for multiple metrics to illustrate how beta diversity is related to abundance and phylogeny: unweighted Unifrac (A), weighted Unifrac (B), and Bray-Curtis (C). Each point represents a unique sample. Coloring indicates individual koalas, shape indicates type of treatment or sample (circles—antibiotic treated, triangles—no antibiotics, squares—samples from the built environment), fill indicates whether the koala lived or died. Numbers in legend indicate the number of samples for that koala. Colors were chosen based on Martin Krzywinski's Color Blindness Palette for improved viewing by those with color blindness.

collection techniques (*Barker et al., 2013*). To collect fecal material from the inside of the fecal pellet, the pellet was placed partway into a sterile 2 ml Eppendorf tube, the pellet was broken in half, and the innermost material from the middle of the pellet was removed with clean, sterile forceps. For each sample, approximately 150 mg of the fecal pellet was transferred with sterile forceps into a new sterile 2 ml Eppendorf tube. DNA was then extracted from this inner-pellet material using the Qiagen QIAamp Fast DNA Stool Mini Kit (following the manufacturer's protocol). DNA was eluted in a final volume of 100 μl in 1.5 ml LoBind Eppendorf tubes and stored at −80 °C.

To confirm that there was bacterial DNA in the samples, we first tested for PCR amplification of 16S rRNA genes using full-length eubacterial 16S rRNA gene PCR primers, 27F (5′-AGAGTTTGATCMTGGCTCAG-3′) and 1492R (5′-TCNGGYTACCTTGTTACGAC

TT-3′). Amplifications were carried out in an Eppendorf Mastercycler® PCR Cycler in 50 µl reactions containing 5 µl of the eluted DNA from each samples' extraction, 18 µl sterile Mill-Q® $H_2O$, 25 µl MyTaq™ HS Mix 2x (Bioline, London, UK), and 1 µl of 0.2 µM concentration of each primer 27F and 1492R (Sigma-Aldrich, St. Louis, MO, USA). The cycling conditions were: (1) 95 °C for 10 min, (2) 35 cycles of 45 s at 95 °C, 1 min at 50 °C, and 2 min 20 s at 72 °C, (3) a final incubation at 72 °C for 7 min and (4) holding at 4 °C upon completion. A negative control was included in every reaction which replaced the 5 µl of DNA with 5 µl of sterile Mill-Q® $H_2O$.

The PCR fragments were visualized by 45 min of 120 V electrophoresis on a 1% agarose gel stained with ethidium bromide. All samples showed a prominent band at the expected length (∼1,465 base pairs) for bacterial 16S rRNA genes; no bands were visible with the negative control of replacing the 5 µl of DNA with 5 µl of $H_2O$.

## Sequencing

PCR amplification of the V4 region of the 16S rRNA gene was performed using primers 515 F and 806 R, as recommended by Caporaso et al. and modified by the addition of a custom barcode system described previously by *Lang, Eisen & Zivkovic (2014)* and *Caporaso et al. (2012)*. In addition to primers, Invitrogen Platinum SuperMix was also used to perform PCR on 5 ng of DNA for each sample following previously established protocols (*Lang, Eisen & Zivkovic, 2014*). PCR cycling conditions were (1) 95 °C for 2 min, (2) 30 cycles of 95 °C for 30 s, 55 °C for 30 s, 72 °C for 1 min, and (3) 72 °C for 3 min. PCR cleanup and normalization was performed according the manufacturer protocol for the Invitrogen 96 well SequalPrep Normalization Plate. The resulting DNA elution of all pooled samples was further purified and concentrated according to the manufacturer protocol for the NucleoSpin® Gel and PCR Clean-up Kit (Macherey-Nagel, Duren, Germany). All DNA was quantified with Qubit® dsDNA HS Assay Kit (Thermo Fisher Scientific, Waltham, MA, USA). Purified DNA of a final concentration of 34.1 ng/µl was submitted for sequencing at the UC Davis Genome DNA Technologies Core in a multiplexed Illumina MiSeq lane. We generated 8,889,513 250 bp pairs of raw reads and 7,020,375 253 bp merged reads, using a custom script (available in Github: https://github.com/gjospin/scripts/blob/master/Demul_trim_prep.pl) to assign each pair of reads from the custom dual barcode system to individual samples. All subsequent analysis of sequences was done on the merged reads.

## Data analysis

Sequences were analyzed using the QIIME (Quantitative Insights Into Microbial Ecology) version 1.9.1 workflow (*Caporaso et al., 2010*). Quality filtering and chimera removal were performed on the sequencing reads before downstream analysis. Operational taxonomic units (OTUs) were picked at 97% similarity with QIIME's script (pick_otus_through_otu_table.py) and clustered using the UCLUST algorithm (*Edgar, 2010*). Taxonomy was assigned using the taxonomy assignment script in QIIME (assign_taxonomy.py) with the open reference database, BLAST and reads that were not assigned a taxonomy via that approach were assigned using the RDP Classifier (*Altschul et*

*al., 1990*; *Wang et al., 2007*). OTUs with five or fewer sequence representatives were filtered out. Additionally, the data was rarefied to 5,000 reads per sample. OTUs that did not have a taxonomic assignment at the 'order' level were filtered out.

Analysis was performed on samples from 11 individual koalas, nine of which had been treated with antibiotics and two that had not. Samples were collected every three to four days from the day the koala was first administered antibiotics until they were released or deceased; there were different numbers of samples collected from individual koalas because individual treatment time was varied. To normalize the analysis, only seven samples were used from each individual: the first sample, the last sample, and five samples that were selected to provide roughly equal chronological spacing from the treatment period. To ensure we only included OTUs capable of conveying group-level effects, we filtered the relative OTUs abundance table to only include OTUs that are present in two or more samples. All OTUs that did not satisfy this threshold were grouped together (sum of their abundances per sample) so that the underlying distribution of OTU abundances across samples did not change. This resulted in 1,511 OTUs across 77 samples. All negative controls for the DNA extraction kit, PCR, and sequencing were below the minimum 5,000 counts per sample that we set (see above) and thus were excluded from downstream analysis. Koalas R, P, and J were treated with a second antibiotic in addition to the same antibiotic as koalas A, B, D, E, F and G. However, calculations using the software R (*R Core Team, 2013*) determined no statistical difference existed between these two treatment groups (see 'Results') so they were pooled into a single antibiotic group.

## Identifying predictive OTUs of fate: random forest analysis

To test whether a given set of OTUs was predictive of koala fate, and thus identify correlations between survival and intestinal microbiome composition, we applied the Random Forest model (available in Python's scikit-learn package (*Pedregosa et al., 2011*)), a supervised machine-learning technique, to our dataset. In our model, we assigned OTUs as the features and identified the set of OTUs that can more accurately predict the fate (i.e., lived or died after treatment) of each individual. Using the Random Forest model, we examined the microbiome of all of the individuals in the antibiotic treatment group and determined the set of OTUs that create the highest distinction among the individuals that survived and the individuals that died.

The goal of the Random Forest classifier is to learn dependencies and complex relationships (both linear and nonlinear) among the features (here, relative OTU abundances) and find a set of features that are the most discriminatory among the groups and can improve the predictive accuracy of the model. It is important to note that the set of OTUs found in Random Forest model can be a combination of both abundant and relatively rare OTUs across samples.

Finally, there is an importance score assigned to each OTU that shows how predictive this OTU is in classifying the output. We measured the success of the Random Forest model by $k$-Fold cross-validation; this includes training the model on a subset of samples and then using the patterns learned to classify the remaining samples that were not used in the training step.

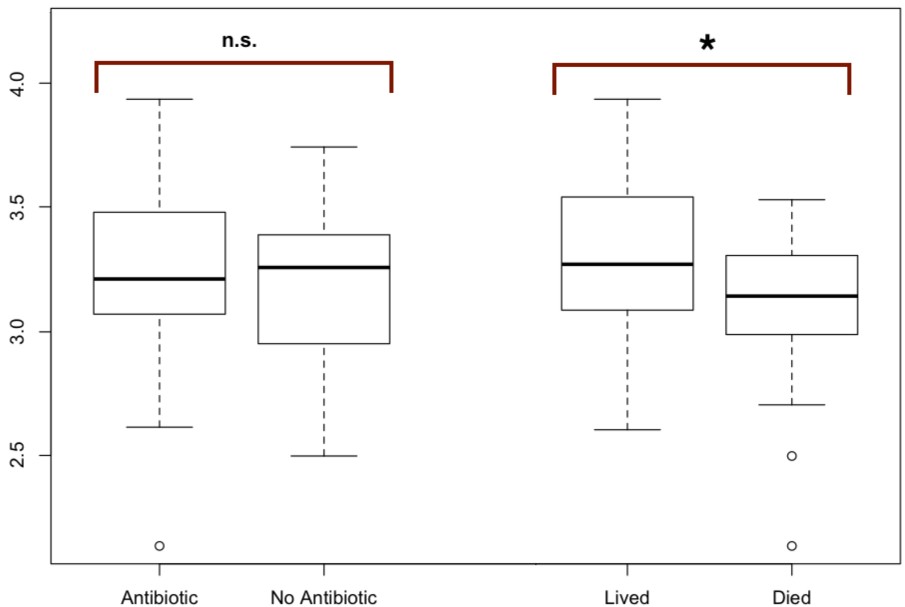

**Figure 2 Modified boxplots of alpha diversity (Shannon index) for microbial communities found in koala fecal samples based on analysis 16S rRNA gene sequences.** Data is presented for pools from different groups of koalas. The Shannon diversity index was calculated for each sample using R and averages for each group of interest were then calculated. The average alpha diversity for each sample was not significantly different between samples from koalas treated with antibiotics ($t$ (test statistic) = 1.1413, $df$ (degrees of freedom) = 52.692, $p$-value = 0.2589). The average alpha diversity for samples from koalas that lived through antibiotic treatment was found to be significantly greater than the average alpha diversity for samples from koalas that died during their admission which included antibiotic treatment ($t$ = 2.9239, $df$ = 38.43, $p$ = 0.005768). Outliers (shown by open circles) represent values that are 1.5 times greater than the difference between the third and first quartiles of the data set.

## Diversity analysis

For the purpose of tracking antibiotic effects on the intestinal microbiome diversity, the mean alpha diversity of each sample was measured by the Shannon Diversity Index using the vegan package in R to measure variation within samples (*R Core Team, 2013*; *Oksanen et al., 2013*) (Fig. 2). Statistical significance of the differences in alpha diversity between groups was calculated using a Welch two-sample $t$-test in R. To determine variations in microbial communities between samples, principal-coordinate analysis (PCoA) plots were generated for weighted UniFrac, unweighted UniFrac, and Bray-Curtis distances using the QIIME software (*Caporaso et al., 2011*). Statistical significance of the differences between groups was calculated by either PERMANOVA or ANOSIM with 9,999 permutations in QIIME using the compare_categories.py script (*Caporaso et al., 2010*).

## OTU co-occurrence network analysis

We investigated the potential interactions among various microbial taxa using network analysis. It has been shown that studying OTU co-occurrences patterns using network analysis provides insight into dynamics of complex microbial systems (*Barberán et al., 2012*; *Faust et al., 2015*; *Faust et al., 2012*). Here, we built a network of the OTUs based on

the presence/absence patterns of OTUs across koala samples using the first and the last sample of individuals. Using our OTU table, each sample can be defined as a binary vector of OTUs showing presence/absence of OTUs and thus each OTU can be defined as a binary vector of that OTU presence/absence patterns across samples. To obtain co-occurrence patterns between every pair of OTUs and thus building the OTU co-occurrences matrix, we used the dot product of every two OTU vectors. The co-occurrence matrix was further used to build the network.

Each OTU represents a node in the network and there exists an edge between every two nodes if they have co-occurred together. The edge weights are the dot product of each OTU vector pair and are representative of how many times those two OTUs co-occurred together. We built a total of three OTU co-occurrence networks for the following sample subsets: (1) The initial time points for all individuals that were subjected to antibiotic treatment regardless of fate; (2) the final time point for individuals who lived; and (3) the final time point for individuals who died. Note that since we have smaller number of individuals in the deceased group, we expect to observe more nodes in the network built from this group because it is more likely to see an OTU present in three samples than it is to observe it in six individuals (case of released group) or nine individuals (network built based on the initial time point of all individuals). Once the networks were built, we obtained clusters of OTUs that co-occurred together in each of these networks separately. We looked at the differences among clusters of OTUs co-occurring in the initial sample and compared that to OTU clusters that co-occurred in the last sampling. We then identified the intersection of the co-occurring OTUs among these three networks.

## RESULTS

Using the QIIME (Quantitative Insights Into Microbial Ecology) workflow, taxonomic assignments were made for 5,934 OTUs (See 'Methods'). After rarefaction to 5,000 reads per sample, the resulting OTU table consisted of 156 samples representative of various time-points from 11 individual koalas. We removed OTUs that did not have any taxonomic assignment at the 'order' level. Finally, we selected seven samples from different time-points from each individual, including the sample collected the day each koala was to begin treatment and the last sample collected before the individual was released or deceased for subsequent analysis.

To test for significant differences between metadata groups, Welch two-sample $t$-tests of the mean alpha diversity were performed in R. Of these tests, we found no significant difference in koala hospital location ($t = 0.544$, $df = 77.0$, $p = 0.587$), koala sex ($t = 1.81$, $df = 86.2$, $p = 0.073$), antibiotic treatment regime ($t = 1.84$, $df = 52.9$, $p = 0.071$), or whether the koala was administered antibiotics ($t = 1.1413$, $df = 52.692$, $p$-value = 0.2589). However, we did find whether a koala lived through treatment ($t = 2.9239$, $df = 38.43$, $p = 0.005768$) to be significant.

### Diversity analysis

Diversity analysis was performed on samples all time points for every koala as outlined in Table S1, with outliers removed where applicable, unless otherwise stated.
### Alpha diversity: antibiotic vs. no antibiotic

The average alpha diversity for each sample, as calculated by the Shannon Diversity index, was not significantly different between samples from koalas treated with antibiotics compared to samples from koalas that were not administered antibiotics ($t = 1.1413$, $df = 52.692$, $p = 0.2589$).

### Alpha diversity: antibiotic and fate

The average alpha diversity for samples from koalas that lived through antibiotic treatment was found to be significantly greater than the average alpha diversity for samples from koalas that died over the course of their admission, which included antibiotic treatment ($t = 2.9239$, $df = 38.43$, $p = 0.005768$) (Fig. 2).

### Beta diversity: lived vs. died

PCoA plots including all of the samples collected were generated for three different distance-based calculation methods: Bray-Curtis, weighted Unifrac, and unweighted Unifrac (Fig. 1). We found statistically significant differences in average diversity between samples from antibiotic-treated koalas that lived to samples from antibiotic-treated koalas that died for two of the three ordination methods used: Bray Curtis ($p = 0.0017$), unweighted UniFrac ($p = 0.0105$). Weighted UniFrac was found to be not significant ($p = 0.1507$).

### Beta diversity: lived vs control

For samples from koalas on antibiotics that lived to samples from koalas not on antibiotics, differences in average diversity was found to be statistically significant for Bray Curtis ($p = 0.0253$), unweighted UniFrac ($p = 0.0002$), but not significant for weighted UniFrac ($p = 0.2537$).

### Beta diversity: died vs control

The difference in average diversity between samples from koalas on antibiotics that died to samples from koalas not on antibiotics was found to be statistically significant for all three ordination methods used: Bray Curtis ($p = 0.0001$), unweighted UniFrac ($p = 0.0001$), and weighted UniFrac ($p = 0.0001$).

## Random forest analysis

Using the Random Forest model with a 3-fold cross-validation, we determined that individuals in the antibiotic treatment group were classified accurately into two groups, those that survived and those that died, with an accuracy of 92 percent and a total of 516 predictive OTUs. The OTU that was most predictive of fate was classified (using the QIIME taxonomy assignment script (assign_taxonomy.py) (*Caporaso et al., 2010*)) as *Lonepinella (sp. koalarum)*. The distribution of the feature importance was scored with the gini impurity criterion (measures the randomness of false label assignment). Using this distribution, we identified the most predictive top 20 percent OTUs and their distribution across samples (Fig. 3).

## OTU co-occurrence network analysis

The resulting bacterial networks for co-occurring OTUs all had a density of one, meaning that there existed an edge between every two nodes (see 'Methods'). Because all three

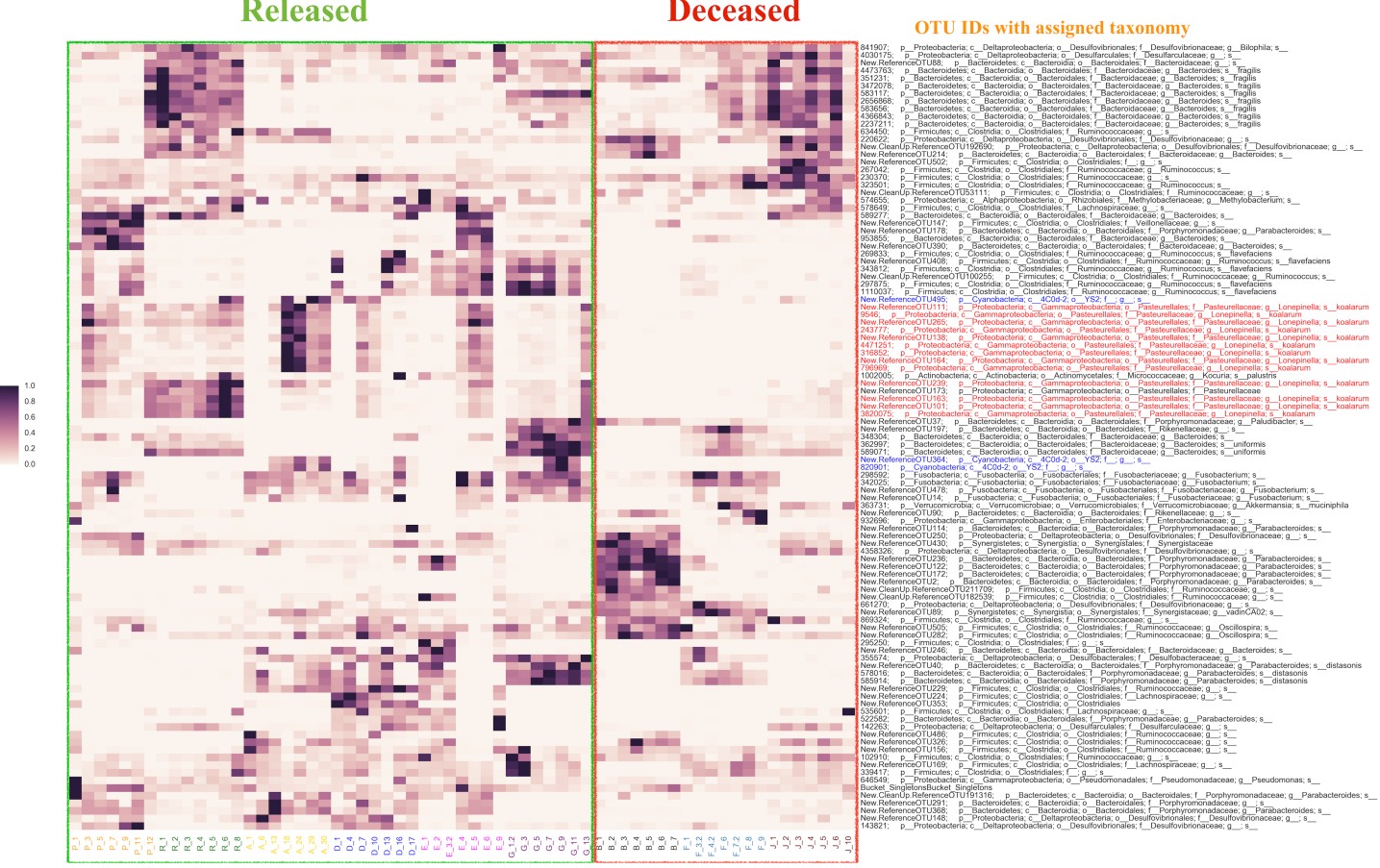

**Figure 3** **The distribution of the top 20% of feature-importance OTUs across individual samples as determined by Random Forest Analysis.**
Individual samples on the *X* axis are organized by the individual koalas from which they came (represented by unique colors), and divided by 'Released' and 'Deceased'. The *Y* axis represents the taxonomic assignment (see 'Methods') for each of the top 20% OTUs that were the most predictive of fate according to our Random Forest Analysis. The coloring of the text for the OTU names is used to highlight specific taxa of interest flagged by network analysis (see main text). Highlighted in red are OTUs identified as *Lonepinella koalarum* that the network analysis identified as the most predictive OTUs of fate. Highlighted in blue are OTUs identified as 'Cyanobacteria YS2', which were identified as being highly predictive of fate and are of interest because they are Melainabacteria. The density of each point in the heatmap is representative of the relative abundance of each OTU for each sample.

networks (antibiotic treated koalas ($N = 9$), koalas that lived ($N = 6$), and koalas that died ($N = 3$)) had a density of one, they were defined as cliques and therefore considered to be one cluster. In more detail, the network that was built based on the initial time point from nine individuals resulted in 34 nodes with 561 edges. The average weighted degree was 297. Similarly, for the network built for the final time point for three individuals that were deceased, there were 100 nodes with 4,950 edges with an average weighted degree of 297. For the network of six released individuals, there were 55 nodes with 1,485 edges and average weighted degree of 324. As a final step, we looked at the intersection of co-occurring OTUs among the networks we built. Twenty-four of the OTUs that were co-occurring at the initial time point still existed in all of the samples in both deceased and released individuals at the final time point. There were four OTUs we determined to be

of most interest, New.ReferenceOTU131, New.ReferenceOTU23, New.ReferenceOTU271 and New.ReferenceOTU265, which existed in all samples at initial time point and existed in all released samples. However, they were absent from several samples of koalas that died. We note, the taxonomic identification suggests that New.ReferenceOTU265 is most closely related to sequences that were annotated as being in the *Lonepinella genus.* For more information on the OTUs that resulted from network analysis, see Table S1.

## DISCUSSION

While antibiotics have many benefits in terms of treating organisms with bacterial infections, they also can cause a disturbance to the microbial communities of the host. Koalas are frequently administered antibiotics for extremely prevalent *Chlamydia* infections, but the effect of antibiotic treatments on their intestinal microbial communities has not been investigated to date. We believe this is likely to be especially important for koalas, particularly if, as discussed above, they are dependent on the microbes in their intestinal to break down the toxic components of their eucalyptus diet. This study characterized the intestinal microbial community of 11 koalas over time until they were released or deceased, nine of which were administered antibiotics as treatment for *Chlamydia* infections. Of the metadata collected (e.g., koala hospital, sex, etc.), we found that whether an antibiotic-treated koala lived or died to be the only parameter that was significantly correlated with patterns in our analysis. Therefore, our paper focuses on the differences between koala treated with antibiotics that lived and koalas treated with antibiotics that died.

### Diversity analysis
#### *Comparisons between antibiotic treated and untreated individuals*
Several studies have shown antibiotic treatment can reduce alpha diversity of mammalian intestinal microbiota (*Knecht et al., 2014*; *Croswell et al., 2009*). However we did not observe a significant difference in alpha diversity between samples from koalas that were administered antibiotics compared to samples from koalas that were not exposed to antibiotics (Fig. 2). This result indicates that antibiotic treatment may have caused compositional shifts within the intestinal microbiome, without affecting overall species diversity. While the lack of significant changes in alpha diversity suggests that antibiotics have a subtle effect on the microbiome, there are also other putative explanations for this lack of a difference in alpha diversity in treated and untreated koalas, including: (1) low doses of antibiotics used here. There could be many explanations for this such as the antibiotic being metabolized quickly and therefore not having as strong of an impact on intestinal microbiota, for example; (2) possible lack of an effect of these antibiotics on the microbial community composition in a way that would be reflected alpha diversity analysis; (3) possible effects of captivity (i.e., transitional stress, environmental changes, etc.) which may mask some of the effects of antibiotics (at least on alpha diversity). We note, there are still differences in community composition in the antibiotic treated vs. untreated samples—the lack of any significant difference is only seen regarding alpha diversity.

We did not find a statistically significant difference in alpha diversity between the two antibiotic treatment regimes used. It is not surprising that the two different antibiotic treatment methods (Chloramphenicol vs. Enrofloxacin and Chloramphenicol) did not have statistically different effects on the intestinal microbial community of koalas in part because both are broad-spectrum antibiotics.

### Comparisons between antibiotic treated koalas that lived and those that did not

We did find a statistically significant difference in the alpha diversity between samples from koalas that were administered antibiotics that lived and samples from koalas that were administered antibiotics and died. This result is consistent with our hypothesis that a higher richness and evenness in the initial intestinal microbial community composition may be more important to koala health during antibiotic treatment than the direct impact of the antibiotic treatment itself.

We examined the beta diversity of different treatment groups (antibiotics and lived, antibiotics and died, no antibiotics) in a pairwise manner using three different metrics: Unweighted Unifrac, weighted Unifrac, and Bray-Curtis. The Unweighted Unifrac metric measures the dissimilarities in phylogenetic distances of OTUs that are present or absent from samples, while the weighted UniFrac metric measures the dissimilarities in phylogenetic distances of OTUs that are present or absent from samples weighted by the abundance of those OTUs. The Bray-Curtis metric takes into account the abundance of present or absent OTUs independent of phylogeny. The differences in microbial communities between samples from koalas on antibiotics that lived versus samples from koalas on antibiotics that died were found to be statistically significant for the Bray Curtis and unweighted Unifrac metrics but not for weighted Unifrac. We interpret these results as showing that the there are differences in both the abundance of specific OTUs (as measured by the Bray-Curtis metric) and phylogenetic diversity of OTUs (as measured by unweighted Unifrac). We are not certain why the weighted Unifrac analysis did not show significant differences but we note that others have reported seeing significant differences in communities for Bray-Curtis and weighted UniFrac metrics but not for unweighted UniFrac (*Lozupone & Knight, 2005*; *Lozupone & Knight, 2015*). This may be related to the method of randomization used in the weighted UniFrac calculation for which there is some debate (*Lozupone & Knight, 2015*; *Long et al., 2014*).

We also compared the samples from koalas on antibiotics that lived and samples from koalas on antibiotics that died to samples from koalas that were never administered antibiotics. We found the difference in average diversity between samples from koalas on antibiotics that lived to samples from koalas not on antibiotics to be significant for Bray Curtis and unweighted UniFrac metrics, but not significant for weighted UniFrac. The difference in average diversity between samples from koalas on antibiotics that died to samples from koalas not on antibiotics was found to be statistically significant for all three ordination methods used: Bray Curtis, unweighted UniFrac, and weighted UniFrac. These results indicate that the structure of the overall communities are different between koalas that died on antibiotic treatment compared to koalas that survived antibiotic treatment or

were in the control group. These beta diversity results are consistent with the alpha diversity results discussed above. One possible explanation for this is that the initial community composition and structure are important to surviving antibiotic treatment. However, we are not able to rule out other possible explanations with the data available at this time.

## Random forest and co-occurrence network analysis

To identify the most predictive OTUs correlated with whether or not a koala survived antibiotic treatment, we performed a Random Forest analysis of OTU relative abundance and presence/absence. The Random Forest analysis revealed that an OTU identified as *Lonepinella koalarum* is the most predictive OTU of whether or not a koala lived or died following the administration of antibiotics. Koalas that died after antibiotic treatment had much lower relative abundance (sometimes even zero), of this *L. koalarum* OTU compared to koalas that survived antibiotic treatment (Fig. 3). We believe that this correlation of the abundance of *L. koalarum* in koala intenstines with surviving antibiotic treatment is potentially a key finding, as *L. koalarum* is known to be a tannin-degrading microbe in the koala intestinal (*Osawa et al., 1995*). Our finding that the set of OTUs predictive of koala survival classified with an accuracy of 92% suggests there is a strong correlation between intestinal microbiome composition and koala prognosis, which is consistent with our initial hypothesis.

In addition to the Random Forest analysis, an OTU co-occurrence network analysis also revealed that *L. koalarum* was correlated with whether or not koalas lived after antibiotic treatment. *L. koalarum* was one of only four OTUs that was present at the beginning and at the end of antibiotic treatment in koalas that lived, but absent in at least one koala that died after antibiotic treatment.

Another finding from our Random Forest analysis was that the OTU "Cyanobacteria_YS2" was in the top 30 most predictive of whether or not a koala lived following antibiotic treatment. Recently, it was shown that the group to which this OTU was assigned to is actually Melainabacteria, a new phylum of non-photosynthetic Cyanobacteria that has been found in numerous mammalian intestines (*Soo et al., 2014*; *Di Rienzi et al., 2013*).

Overall, the results of Random Forest and co-occurrence network analyses support our hypothesis that the administration of antibiotics, regardless of the combination of the treatment, was associated with a change in the presence and relative abundance of *L. koalarum,* a known tannin-degrader. This is not only important for veterinarians to consider when administering antibiotics to koalas, but also for the development of *Chlamydia* infection treatments that do not impact this critical intestinal microbe.

Potential confounding variables that may have influenced our results are sample storage (*Lauber et al., 2010*), DNA extraction method (*Salter et al., 2014*), PCR (*Brooks et al., 2015*; *Brown et al., 2015*), and sequencing (*Schirmer et al., 2015*). Due to not having full medical histories of the koalas in this study, it is also possible that unknown confounding variables (e.g., immune status or prior environment exposures), rather than antibiotics, contributed to our results. Other environmental variables that may be confounding factors that we were unable to test for include the effects of being handled by hospital staff, koala

compliance with antibiotic treatment, differences between wildlife hospitals' environments and procedures, and day-to-day care.

## Conclusions

Our findings are consistent with numerous other papers that suggest antibiotic treatments can cause a disturbance to intestinal microbial communities e.g., *Jakobsson et al., 2010*; *Dethlefsen et al., 2008*; *Jernberg et al., 2007*; *Sullivan, Edlund & Nord, 2001*. Such disturbances are likely to be particularly important in species like koalas, where it is thought that the intestinal microbial community may be required for survival (in this case, detoxifying their food). This is in contrast to many other animals where the intestinal microbiome is important but not necessarily essential for survival. In addition to showing differences in antibiotic treated and untreated koalas, we also found differences in richness, evenness, and structure of intestinal microbial communities in antibiotic treated koalas for those that survived versus those that did not. In particular, we found that the relative abundance of some key OTUs are correlated with survival. This is consistent with our hypothesis that koala survival after antibiotic treatment is related to whether or not key OTUs persist after the antibiotic treatment. We believe this suggests that there may be a need to develop alternative treatments for koala *Chlamydia* infections without the use of antibiotics and/or supplementing the koala diet with probiotics during antibiotic treatment. Furthermore, our conclusions may be transferable to other species that consume high-tannin diets. Future studies about this topic could be more comprehensive in scale (e.g., sample size) and depth (e.g., metagenome rather than 16S rRNA gene sequencing). It would be valuable to track koalas over a longer period of time rather than just their time in captivity. Furthermore, an important area to investigate is the role of pap consumption in the colonization and microbial community structure changes of koala intestines. Given our findings, it would be particularly important to investigate the pap of antibiotic treated mother koalas and the impacts this has on joey health.

## ACKNOWLEDGEMENTS

We would also like to thank Cheyne Flanagan (Port Macquarie Koala Hospital), Rosie Booth and Amber Gillet (Australia Zoo Wildlife Hospital) for coordinating and help with sampling the koalas in the study. Special thanks to Alyce Taylor-Brown, Cassie Ettinger (ORCID 0000-0001-7334-403X) and David Coil (ORCID 0000-0001-6049-8240) for their guidance and expertise throughout different stages of this project. Thank you to Guillaume Jospin (ORCID 0000-0002-8746-2632) for providing his bioinformatics skills. And lastly, thank you to the Bosward family for their help with logistics.

### Funding

This work was funded by the Alfred P. Sloan Foundation as part of their ''Microbiology of the Built Environment'' program. This work was also funded by an Indiegogo crowdfunding

campaign. The funders had no role in study design, data collection and analysis, decision to publish, or preparation of the manuscript.

### Grant Disclosures
The following grant information was disclosed by the authors:
Alfred P. Sloan Foundation.
Indiegogo crowdfunding campaign.

### Competing Interests
The authors declare there are no competing interests.

### Author Contributions
- Katherine E. Dahlhausen conceived and designed the experiments, performed the experiments, analyzed the data, contributed reagents/materials/analysis tools, prepared figures and/or tables, authored or reviewed drafts of the paper, approved the final draft.
- Ladan Doroud analyzed the data, prepared figures and/or tables, authored or reviewed drafts of the paper, approved the final draft.
- Alana J. Firl analyzed the data, authored or reviewed drafts of the paper, approved the final draft.
- Adam Polkinghorne and Jonathan A. Eisen conceived and designed the experiments, contributed reagents/materials/analysis tools, authored or reviewed drafts of the paper, approved the final draft.

### Animal Ethics
The following information was supplied relating to ethical approvals (i.e., approving body and any reference numbers):

Sample collection from the Australia Zoo Wildlife Hospital and Port Macquarie Koala Hospital qualified for exemption from approval by the University of the Sunshine Coast Ethics Committee.

### DNA Deposition
The following information was supplied regarding the deposition of DNA sequences:

Figshare: https://doi.org/10.6084/m9.figshare.5096392.v1.

NCBI Bioproject PRJNA391972, accession numbers SRX2961911–SRX2962051.

### Data Availability
Dahlhausen, Katherine (2017): Koala Project Sequence Raw Data. figshare. DOI 10.6084/m9.figshare.5096392.v1.

### Supplemental Information
Supplemental information for this article can be found online at http://dx.doi.org/10.7717/peerj.4452#supplemental-information.

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
