# Peer review of "Characterization of shifts of koala (Phascolarctos cinereus) intestinal microbial communities associated with antibiotic treatment"

_PeerJ, doi:10.7717/peerj.4452_

## Round 0.1 · original submission · Major Revisions

· Academic Editor

Major Revisions

While all 3 expert reviewers concluded that this is a very interesting and well written manuscript, they also identified a number of issues related to study design, lack of experimental details, and data analysis & interpretation.

Please provide a revised manuscript, where these issues are appropriately addressed.

Reviewer 1 ·

Basic reporting

The study is quite well written. Is the term guts the correct and more scientific term in this work. Background is good and in detail. Quoteing of literature is good.

Experimental design

Overall too many results are not well conclusive as admitted by the authors themselves. The work is very preliminary.
A main weakness of this work is for the analysis of the fecal pellets as samples. If I understand fully then the koala has many parts to its gut that do different parts of the digestion. Is this correct. The fecal pellet is the end result. How can it be properly representative of the part of the gut where key digestion is going on. Can the authors expand their study to show at least a few koala that are analysed with different part of the gut. Or is it already know that the fecal composition is truly representative.

Validity of the findings

Some of the statements the authors make in the discussion do not seem to be correct or supported.
“Low doses of antibiotics were used here”. Is this a true wording. If it is true wording, then why was this done. Did the veterinary staff involve do this.
How would “captivity mask the effects of antibiotics”. The authors should plan to give some explanations.
The koalas were treated with different antibiotics but the authors say that the difference do not matter. They should expand on this claim.
They compare with an animals that was not treated with the antibiotics. But this was only one or perhaps two animals. How can they do just one animal. How did they do statistics analysis with such low number.
How did the animals die. This is very important to explain. Was it due to the chlamydiae infection I guess.
Is the Lonpinellea organism known to be sensitive to the antibiotics. Could the authors test this in an actual agar plate assay. Would be very good to show confirmation.
Also is the word guts sufficiently scientific description of the koala intestine parts.

Reviewer 2 ·

Basic reporting

The information about pap consumption is very interesting and would be a fascinating microbiome study on its own. (Pre/during/post pap consumption microbiome along with the microbiome of the pap sample itself.) That said, the first paragraph of the intro, while interesting, seems out of place. Pap is never mentioned again in the entire paper. I suggest either finding a way to relate pap to the results or drop the information on pap and instead focus the intro on the importance of evaluating the microbiome in a vulnerable (IUCN) species that depends on the microbiome for degradation / detoxification of dietary items. (There are also some good creosote/woodrat articles that discuss microbiome and detoxification that would provide additional support in your intro.)

In your introduction of Chlamydia, including a sentence or two about transmission would also be helpful – primarily STI? But vertical transmission and transmission through pap also possible??

Figure 2: The other piece of information that would be quite useful in this figure is wildlife hospital and antibiotic protocol. (These could be supplemental figures, or you could comment about clustering or lack of clustering by hospital / antibiotic protocol.) This figure would also be more readable if you used shapes and open/closed symbols to differentiate things like antibiotic/control and released / deceased. (You can edit PCoA’s produced in QIIME by opening the “_pc.txt” files in Excel and scatter plotting the first 2 columns (which represent PC1 and PC2). There are ways to do this in R as well.

Lines 351-352: In the intro, your “hypothesis” is: “we hypothesized that the reason for adverse side effects of antibiotics administered to koalas with Chlamydia infections is due to disturbance in the microbial communities present in the koala gastrointestinal tract, especially those microbes responsible for tannin degradation.”
This is not exactly consistent with your discussion statement that reads “This result is consistent with our hypothesis that a higher richness and evenness in the initial gut microbial community composition may be more important to koala health during antibiotic treatment than the direct impact of the antibiotic treatment itself.” The hypothesis is mildly overstated at other places in the discussion as well. (The intro hypothesis is very broad. The discussion makes it sounds like the hypothesis was specific.)

Experimental design

Sample Collection: Koalas were not randomized to treatment with or without antibiotics, so how do the wildlife hospitals decide which koalas to treat with antibiotics? (Does this selection bias your groups? i.e. only the “clinically mild” animals received no antibiotics??)

Can you provide any information regarding the following: Were any of the animals co-housed or did they have contact with each other or shared spaces? What were typical husbandry / sanitation procedures at the hospitals? What were captive animals fed at each hospital?

“Built environment” samples are included in the PCoA but collection methods for these samples are not described. (It was a GREAT idea to collect these samples.. nice work. Out of curiosity, did the built environment differ / cluster by hospital?)

What was the treatment protocol (dose, frequency, duration) and did it vary between individuals?

Table 1: Additional information that would be helpful in this table – wildlife hospital from whence the sample came, dose and concentration of the drug and frequency and duration of treatment. (Number of ml of drug given is not helpful. I’d rather see something like “chloramphenicol, 2 mg/kg, twice a day for 7 days”).

Was the first sample(s) collected prior to antibiotic treatment?

Lines 239-243: So, an OTU can only be a node if it appears in all samples within a group? Please clarify. More typically, I see this analysis with all OTUs as nodes (regardless of whether they appear in multiple samples or not), and edge weight indicates commonness of co-occurrence within samples. I would expect number of nodes and co-occurrences to be greater with more samples rather than lesser?

Validity of the findings

Lines 295-296: I’m still unclear about the co-occurrence network. There was an edge “between every two nodes” – are you saying every node (OTU) co-occurred with every other node? Or do you mean “an edge connected each node (OTU) to one or more other nodes in the network”

Lines 300-3001: Why were there more nodes (OTUs) in 3 individuals than there were in 9 individuals? (I know you try to explain this above.)

Line 309: Do you mean, “However they were not present in all samples of koalas THAT DIED.” (instead of “that were released.”)

In the Discussion you state that “Of the meta data collected (e.g. weight, age, geographical location, koala hospital, sex, Chlamydia status, etc.), antibiotics and fate (lived or died) were the only parameters that were significantly correlated with patterns in our analysis.” However you do not detail these results / analyses. Can you provide statistical evidence (PERMANOVA / ANOSIM etc.) or show figures to demonstrate that factors like hospital were not significant in your results? Alternately, limit this statement to the results you tested and shared in the results section.
In the discussion you do not need to restate the t / df / p values from your results section. (Do not re-write the results in Discussion)
Line 335 – it is difficult for the reader to appreciate “low doses” when dose / concentration / frequency / duration of medication was not provided.
Line 337-338 – Your point about captivity is a good one. There are multiple studies that support this idea; wild animals that are moved into captivity decrease in alpha diversity. Citing one or more of these studies here will lend credence to your point.
Lines 341-345 – These results were not included in the results section above.
Lines 351-352 – Were your first samples collected after antibiotics were already given? If so, then “initial gut microbial community composition” would already have been altered.
Lines 357-361 – This reads more like methods (or results).
Line 413 – PICRUSt is not mentioned in the methods.
Lines 425-428: Other confounding variables may include – different hospitals, different diet / husbandry / handling at each hospital, varying dose / duration / frequency of antibiotics?
Line 443: This may be a good spot to speculate about the potential benefit of fecal microbial transplants from healthy koalas as a treatment modality.
The network / co-occurrence analysis needs better explanation and likely has more results worth probing. The only thing mentioned is that New.ReferenceOTU265 was “of interest” and is most closely related
to Lonepinella. Did this OTU co-occur with other microbes in the network? It would be important to understand if Lonepinella alone might make a good probiotic or if it needs “friends” that sustain or support its growth in some way… in other words – what microbes co-occur with Lonepinella?

Additional comments

This study examines the gut microbiota of Chlamydia-infected koalas not on antibiotics (n=2), on antibiotics that survived infection (n=6), and on antibiotics that died of infection (n=3). The subject matter is interesting and valuable as Chlamydia is a common and devastating koala disease that is routinely treated with antibiotics. Koalas are also unique in their ability to digest and detoxify eucalyptus leaves and the gut microbiome likely plays a critical role in this process. The sample sizes are small and many variables / possible biases exist (different hospitals, different antibiotic regimens). There are also multiple variables that are not mentioned / described but could be quite important: does diet differ at different hospitals? How does each hospital choose which animals to give antibiotics to? That said, the findings are interesting and despite small sample sizes, many variables, and non-randomized groupings, the findings provide support for the importance of microbes like Lonepinella in koalas. This finding is not novel; however, previous studies have relied on culture of Lonepinella while this study uses 16S sequencing and microbial co-occurrence networks to come to a similar conclusion about the importance of Lonepinella.

My major concerns are around the sample size and variation in groups – much of which is undescribed. Addressing the questions / comments highlighted in the review will be helpful in determining / describing the limitations of the study.

Minor edits are included below:

Line 52: replace “their” with “its” (singular)
Line 53: replace “are” with “as”
Line 235: replace OUT with OTU
Line 245: replace “c-occurring” with “co-occurring”
Figure 1 caption: replace “form” with “from”
Table 1 – replace “chlorampheniocol” with “chloramphenicol” under “Daily Dose” in almost every line.
Supplemental Table 2 caption: replace “OUT Co-occurrence” with “OTU Co-occurrence” – what does this table mean? Do all of these OTUs co-occur in all koala samples in all groups?

·

Basic reporting

This manuscript is well-written and provides substantial background information regarding the topic. The relevant and appropriate literature is cited and the figures and tables are appropriate for this type of study. Overall, this was viewed as a strength of the manuscript, as it does a good job of explaining the particular research problem for non-specialists in the field.

Experimental design

This work presents original primary research that fits within the scope of PeerJ. The research question itself is well defined, meaningful, and fills in an identified knowledge gap. Given previous work on the Koala microbiome, which is in its infancy, this work advances our understanding of these organisms, and a particular issue that is detrimental to wild populations of koalas.

While the work has a lower number of animals, relative to other work on wild populations, this is understandable given the nature of the study. Methods require a bit more description, as there is presented material both in the results and outside of the results that are not described in detail. Specific details regarding this will be described in section (3) of this review.

However, in general, this work is very interesting and advances our understanding of the koala microbiota as it relates to health.

Validity of the findings

In short, this work presents the microbiota of koalas infected with Chlamydia and treated with antibiotics, relative to koalas that are uninfected. Necessarily, the authors looked at koalas from three different hospitals in order to acquire a larger number of animals for the study. The authors collected fecal samples from these koalas every 3 days for a period of 21 days. Two koalas did not survive antibiotic treatment, and their microbiotas were also investigated, in comparison to the other groups. Overall, this is an interesting study that addresses a critical concern in koala biology, given that Chlamydial infections are decimating wild koala populations.

However, I do have some concerns regarding this work that needs to be addressed:

1. In discussion, it appears that two new pieces of data are presented that are not included in the results.

First, the authors note on line 323 that meta data associated with these koalas were assessed, but only antibiotic vs. non-antibiotic groups were found to be significantly different. I did not find any reference to this in the results (where this should be presented), nor is there any reference to these meta data and their analyses in the methods. This should be provided, and the particular model included (i.e. interaction variables, etc), as a natural hypothesis here is that animals from the different hospitals would cluster together based on geography, rearing habits, etc. For example, in Figure 2, if labels to the individual koalas are added according to hospital, would clustering appear according to this factor? Moreover, this data should be presented in the results section, not the discussion.

Second, an entire section is devoted to PiCRUST analysis, which is not described in the methods. Moreover, the results from this analysis were inconclusive, given that many of the taxa identified (e.g. Lonepinella) do not have sequenced genomes. I would recommend removal of this analysis altogether, as it does not add anything to the study.

2. In Figure 2, samples are included described as "build environment" and described as materials in the koala wards. However, I could not find any reference to this anywhere in the main text of the study. Why were these included? Did these samples and their analyses provide any insights into the study itself? If these were used as an outgroup, they should be mentioned. Moreover, there is no mention of these samples and their collection in the methods.

3. To me, an important consideration of this study is the fact that the authors conducted a longitudinal sampling of all animals. However, this study really only considered the impact of antibiotic use on the microbiota of koalas, rather than conducting an analyses of how the gut microbiota is changing over time. This is an important fact given that antibiotic use would be expected to alter the microbiota, especially over time. For example, in any given koala, either treated or not treated with antibiotics, is there a core microbiota that persists across time? What groups of bacteria disappear over time with antibiotic use, and are there significant differences in the respective timeframe for those koalas that recovered vs. perished? Given that 100+ samples are presented in this study, significantly more analyses could be performed in order to provide insights into what occurs in the gut of these koalas as a result of antibiotic use.

Additional comments

Overall, this is an interesting study, but requires substantially more analyses given the data available to the authors. The authors should pay particular attention to ensuring that the methods are complete and that no new results are presented in the discussion. Finally, the authors should consider the longitudinal nature of the study, and capitalize on this data in order to glean insights into how antibiotic use is changing the gut microbiota, as this may perhaps explain why two of the koalas perished after antibiotic treatment.

---

## Round 0.2 · Minor Revisions

· Academic Editor

Minor Revisions

Both reviewers agreed that you have in general taken previous comments and suggestions into account very well, and the now only have a few remaining minor suggestions to further improve the manuscript.

Reviewer 2 ·

Basic reporting

No comments

Experimental design

No comments

Validity of the findings

No comments

Additional comments

The authors have done a good job addressing many reviewer comments, explaining their responses thoughtfully, and providing thorough clarifications throughout the manuscript.

I suggest only the minor revisions below:

Line 48: Change "intestinal" to "intestines"

Line 82: Change "is" to "are"

Line 134: Change "Leave" to "leaf"; changed "sealed" to "sealing it"

Figure 1 caption: Change "Plotsare" to "plots are"

Line 206: Change "does" to "did"

Line 282 and 355: Change "meta data" to "metadata"

Line 284: Change "gender" to "sex"

Line 341: "They were not present in all samples of koalas that were deceased" - change "were deceased" to "died". Also, this is (unintentionally) misleading. This sentence makes it sounds like Lonepinella was not found in any samples that came from the koalas that died. In line 442, this is clarified: Lonepinella is "absent in at least one koala that died after antibiotic treatment." Suggest rewording line 341 so that it becomes clear that Lonepinella was found in some but not all koalas that died.

Line 371: Change "as strong of impacts" to "as strong of an impact"

Line 369-370: Suggest removing the "1) low doses of antibiotics used here. There could be many explanations… for example." Your beta diversity results (Lived vs. Control and Died vs. Control) demonstrate that antibiotics seem to have a clear effect on microbial composition - even though they did not significantly affect alpha diversity. The other 2 putative explanations you list here are stronger and more plausible.

Line 387: Change "passed away" to "died"

Line 450-452: I suggest adding a brief caveat at the end of this paragraph. This study (for ethical reasons) did not allow you to examine koalas that died that did not receive antibiotics. As such, it is possible that the course of the Chlamydia infection itself affects Lonepinalla presence / abundance independent of antibiotics. (That said, this is more of a scientific point than a clinical point, as the treatment for lack of Lonepinalla due either to antibiotics or Chlamydia would still be to help koalas recover their Lonepinalla populations)

·

Basic reporting

no comment

Experimental design

The authors have adequately addressed the concerns raised in my previous review.

Validity of the findings

Given the limitations of the study, the work presents a characterization of the gut community of koalas receiving or not receiving antibiotics. While descriptive in nature, this work will be of value to researchers in the field, and greatly expands our understanding of the koala gut microbiota.

Additional comments

In general, the authors have addressed the reviews and this iteration of the manuscript is significantly improved. I do have a couple of minor suggestions for the authors to consider:

1. Line 95 indicates that import approval was received from the "United States of America Government". This is rather non-specific, and should be addressed. In particular, I am guessing import permits of some kind were obtained from UW Fish and Wildlife Services? I do not believe USDA APHIS controls import of materials from animals like Koalas. If so, a proper import permit ID should be included here.

2. In the materials and methods, it is unclear where specific activities were performed. I am guessing that sequencing was performed at UC Davis, from DNA material extracted in Australia? This should be noted and the specific sequencing facility should be referenced.

3. Lines 339 - 342 - is there a reason for including the phrase "New.Reference"? Why not simply OTU131? This would make for a cleaner reading and presentation of the OTU numbers.

---

## Round 0.3 · accepted · Accept

· Academic Editor

Accept

This is a very interesting paper indeed. Nice and very relevant work. One last remark - I would suggest to replace "flora" in the abstract with "microbiota"